# The Difference in the Prevalence of Metabolic Syndrome According to Meeting Guidelines for Aerobic Physical Activity and Muscle-Strengthening Exercise: A Cross-Sectional Study Performed Using the Korea National Health and Nutrition Examination Survey, 2014–2019

**DOI:** 10.3390/nu14245391

**Published:** 2022-12-19

**Authors:** Du Ho Kwon, Young Gyu Cho, Hyun Ah Park, Ho Seok Koo

**Affiliations:** 1Department of Health Care Management, Aerospace Medical Center, Cheongju 28187, Republic of Korea; 2Department of Family Medicine, Seoul Paik Hospital, Inje University College of Medicine, Seoul 04551, Republic of Korea; 3Department of Internal Medicine, Seoul Paik Hospital, Inje University College of Medicine, Seoul 04551, Republic of Korea

**Keywords:** exercise, metabolic syndrome, obesity, blood pressure, glucose, triglycerides, cholesterol

## Abstract

Physical activity and muscle strengthening are essential for preventing and managing metabolic syndrome. This study was conducted to investigate the relationship between the prevalence of metabolic syndrome and meeting the guidelines for aerobic physical activity (APA), muscle strengthening exercise (MSE), and combined exercise. We used data from 22,467 Koreans aged 40 years or older, who participated in in the Korea National Health and Nutrition Examination Survey (KNHANES) 2014–2019. We used the Global Physical Activity Questionnaire (GPAQ) to measure physical activity and surveyed frequency of MSE through a questionnaire. Metabolic syndrome was defined according to the American heart association and the National Heart, Lung, and Blood Institute. Compared with none exercise group, odds ratios of APA, MSE, and combined exercise group (CEG) on metabolic syndrome prevalence were 0.85 (95% confidence interval (CI), 0.74–0.98), 0.81 (95% CI, 0.67–0.99), and 0.65 (95% CI, 0.54–0.78) among men, respectively. Among women, ORs of APA, MSE, and CEG were 0.83 (95% CI, 0.73–0.93), 0.73 (95% CI, 0.58–0.91), and 0.74 (95% CI, 0.58–0.93), respectively. This study showed that meeting guidelines for APA and MSE was associated with lower prevalence of metabolic syndrome. Furthermore, subjects who met both APA and MSE had the lowest metabolic syndrome prevalence.

## 1. Introduction

Metabolic syndrome, expressed through insulin-resistant diabetes, hypertension, abdominal obesity, and dyslipidemia, is a risk factor for cardiovascular disease and is known as a precursor of several chronic diseases [1,2]. The prevalence of metabolic syndrome has gradually increased worldwide [3,4]. In Korean men, the prevalence of metabolic syndrome increased from 2008 to 2017, whereas that in women stable [5].

It is necessary to actively manage chronic diseases such as metabolic syndrome to prevent cardiovascular disease. We recommend physical activity and exercise to prevent and manage diabetes mellitus and metabolic syndrome [6,7]. According to physical activity guidelines, aerobic physical activity (APA) is recommended at least 150 min/week of moderate-intensity or 75 min/week of high-intensity aerobic activity. Muscle-strengthening exercise (MSE) is recommended for at least 2 days a week [8]. In addition, both APA and MSE are recommended for diabetes [9]. Among Japanese adults, 35.1% of men and 27.4% of women met physical activity guidelines in 2016 [10]. Among Chinese adults, the proportion of meeting physical activity guidelines increased from 17.2% in 2000 to 22.8% in 2014, but it was still low [11]. According to the Korea National Health and Nutrition Examination Survey (KNHANES), Korean adults meeting guidelines for APA decreased from 58.3% in 2014 to 45.6% in 2020. In contrast, the proportion of Korean adults meeting guidelines for MSE has increased from 21.0% in 2014 to 24.7% in 2020. Only 16.9% of Korean adults met guidelines for both APA and MSE in 2020 [12].

Previous studies have examined the effects of physical activity and exercise on managing metabolic syndrome. According to a meta-analysis, aerobic exercise improves metabolic syndrome components reducing fasting plasma glucose (FPG), waist circumference, blood pressure and triglycerides [13]. In addition, muscle-strengthening training reduces waist circumference, glycated hemoglobin (HbA1c) and systolic blood pressure (SBP) in obese and diabetic patients [14]. In the general population, aerobic exercise has been consistently shown to benefit the management of metabolic syndrome components in some studies [15,16]. However, the effects of MSE and combined aerobic and resistance exercise on metabolic syndrome were inconsistent between studies [16]. Moreover, previous studies have mainly focused on Americans or Europeans and the studies on Asian population are insufficient.

Despite the Korean Health Policy to improve physical activity rate [17], the proportion of meeting exercise guidelines was still low and the prevalence of metabolic syndrome has increased. There are not enough studies about effect of MSE or combined exercise. This study was conducted to investigate the relationship between the prevalence of metabolic syndrome and meeting the guidelines for APA, MSE, and combined exercise using a nationally representative sample of Korean population.

## 2. Materials and Methods

### 2.1. Subjects

We used data from KNHANES, 2014–2019. A two-stage stratified cluster sampling designed the KNHANES to represent the Korean population. Data consists of the health interview survey (medical history, household survey, socioeconomic status, and health behavior survey) and health examination (anthropometric measurement, physical examination, blood pressure, biochemical measurement).

Among 47,309 subjects who participated in KNHANES, 2014–2019, we selected the study subjects as adults over 40 years of age since the prevalence of metabolic syndrome increased in subjects over 40 years [18]. Subjects with missing data on the physical activity questionnaires, physical measurements, high-density lipoprotein cholesterol (HDL-C), triglyceride or FPG were excluded. In addition, we excluded subjects with medical history of stroke, angina, or myocardial infarction, pregnant women, and subjects who had their blood drawn without fasting for more than 8 h. Finally, 22,467 Korean adults (9670 men and 12,797 women) were included for the study analysis. There was no statistical significant difference in the prevalence of metabolic syndrome and the components of metabolic syndrome between subjects who were excluded from the study and those who were included. The study protocol was approved by institutional review board at the Aerospace Medical Center, Republic of Korea Air Force (No. 136367-202202-HR-02-00).

### 2.2. Definition of Metabolic Syndrome and Metabolic Syndrome Components

Trained medical staffs measured waist circumference at the midpoint between the inferior margin of the last rib and the iliac crest, according to World Health Organization guidelines. The participant’s blood pressure was measured three times, after they had been quietly seated for 10 min, and the average of the second and third measurements was used. After fasting for 8 h or more, blood samples were taken to measure total cholesterol, triglycerides, HDL-C, and FPG. We defined hypertension as systolic blood pressure (SBP) ≥ 140 mmHg, diastolic blood pressure (DBP) ≥ 90 mmHg, or taking anti-hypertensive drugs diabetes as FPG ≥ 126 mg/dL, a glycated hemoglobin level ≥ 6.5%, taking oral diabetes medications or using insulin therapy and hypercholesterolemia as total cholesterol ≥240 mg/dL or taking lipid-lowering medications.

We defined metabolic syndrome according to the American heart association and the National Heart, Lung, and Blood Institute [19] and the diagnostic criteria of abdominal obesity for Koreans [20]. Metabolic syndrome was diagnosed as presenting with three or more of the following criteria: (1) high waist circumference (≥90 cm for men or ≥85 cm for women), (2) elevated triglycerides (≥150 mg/dL), (3) low HDL-C (<40 mg/dL for men or 50 mg/dL for women), (4) elevated FPG (≥100 mg/dL) or a diabetic patient, and (5) high blood pressure (SBP ≥ 130 mmHg or DBP ≥ 85 mmHg) or a hypertensive patient.

### 2.3. Assessment of Physical Activity and Exercise Guidelines

We used the Global Physical Activity Questionnaire (GPAQ) to measure physical activity. GPAQ comprises 16 questions to survey various physical activities such as work, transportation, leisure, and recreation. Through the questionnaire, we surveyed whether study participants were active, how many days per week and how many minutes per day were spent on moderate or high-intensity leisure or physical activity for at least 10 min over the past week. These questions divided physical activity into five categories: high-intensity work, moderate-intensity work, moving to a location, high-intensity recreation, and moderate-intensity recreation.

We surveyed frequency of MSE through a questionnaire. For example, have you done sit-ups, push-ups, pull-ups, barbells or dumbbells in the past week? Weekly frequencies ranging from “not at all” to “more than 5 days” were used.

The Korean guideline for physical activity adopts the same basic principles with the US guideline. Recently, the US guideline was revised. According to the US guideline, APA is recommended at least 150–300 min/week of moderate-intensity or 75–150 min/week of high-intensity, and MSE is recommended for at least 2 days/week [9]. The study participants were categorized into four groups; (1) none exercise group (NEG, meeting neither APA and MSE recommendation), (2) APA group (meeting only recommendation for APA), (3) MSE group (meeting only recommendation for MSE), and (4) combined exercise group (CEG, meeting both APA and MSE).

### 2.4. Covariates

We selected covariates according to two previous studies [21,22]. Weight and height were measured up to 0.1 kg and 0.1 cm, and body mass index (BMI) was calculated by dividing weight (kg) by the square of height (m^2^). Smoking history was divided into current smoking and non-smoking, and alcohol consumption was categorized as none (1 or less per year) mild (less than 1 per month), or heavy (2 per month or more). The locality of residence was divided into urban and rural, and household equivalent income was categorized into quartiles. Educational level was divided into 4 categories; elementary school or lower, middle school, high school, and college or higher graduates.

### 2.5. Statistical Analyses

We performed statistical analyses through the R program 5.4.3 (R foundation, Vienna, Austria) using the Survey library and the Gtsummary library. We incorporated sampling weight considering the multistage probability sampling design and the non-responses to be representative of the Korean population. Since 6 years of data were integrated, a weight of 1/6 was applied to each analysis year.

The general characteristics of the study subjects were presented as a mean and standard deviation (SD) for continuous variables through a t-test, and were presented as percentage (%) and standard error (SE) for categorical variables through the chi-squared test.

We used multivariate logistic regression models to estimate the difference in the prevalence of metabolic syndrome according to meeting guidelines for APA and MSE. The NEG was considered as reference group. All analyses were stratified according to sex. The odds ratios (ORs) and 95% confidence intervals (CIs) for the prevalence of metabolic syndrome were measured after adjusting for age in model 1, adjusting for age, alcohol consumption, smoking, and BMI in model 2, and adjusting for age, alcohol consumption, smoking, BMI, education level, household income, and residence locality in model 3. In addition, we used general linear regression models to estimate waist circumference, triglycerides, SBP, DBP, HDL-C, and FPG. Statistical significance was considered using a two-sided *p*-value < 0.05.

## 3. Results

### 3.1. Baseline Characteristics

Among 22,467 subjects, 43.0% were men. The mean age of men and women was 55.3 ± 10.5 years and 56.6 ± 11.2 years, respectively. Men had a higher prevalence of metabolic syndrome than women (33.0 ± 0.6% vs. 24.8 ± 0.5%, *p* < 0.001). The proportions of each exercise group were 42.4 ± 0.6% for NEG, 10.6 ± 0.4% for MSE, 29.7 ± 0.6% for APA and 17.3 ± 0.5% for CEG in men and 52.6 ± 0.6% for NEG, 5.7 ± 0.3% for MSE, 33.5 ± 0.5% for APA and 8.2 ± 0.3% for CEG in women, respectively (Table 1).

Among women, all other exercise groups were likely to have lower BMI, waist circumference, triglyceride, SBP, FPG and higher HDL-C than NEG. (all *p* < 0.001 except *p* = 0.001 for BMI in APA) Among men, CEG was likely to have lower waist circumference and triglyceride and higher HDL-C compared to NEG. (all *p* < 0.001) In addition, APA was likely to have a higher HDL-C (*p* < 0.001), and MSE was likely to have a lower triglyceride (*p* = 0.007) (Table 2).

### 3.2. Metabolic Syndrome Prevalence (Table 3)

The prevalence of metabolic syndrome was significantly lower in all other exercise groups than NEG in both sexes. Among men, CEG had a lower proportion of waist circumference, triglyceride, HDL-C and FPG meeting metabolic syndrome criteria in than NEG (all *p* < 0.001). APA had a lower proportion of low HDL-C category and high FPG category and MSE had a lower proportion of elevated triglycerides category and low HDL-C category. Among women, all other exercise groups had a lower proportion of all metabolic components compared to NEG.

**Table 3 nutrients-14-05391-t003:** The prevalence of metabolic syndrome and its components according to exercise groups.

	Men				Women			
	NEG (Ref)	APA	MSE	CEG	NEG (Ref)	APA	MSE	CEG
Waist Circumference meeting for Mets	36.5 (1.2)%	33.5 (1.2)%	35.4 (1.7)%	29.7 (1.7)% ***	34.0 (0.8)%	22.5 (0.9)% ***	27.2 (1.7)% ***	16.5 (1.7)% ***
Triglyceride meeting for Mets	45.1 (1.2)%	39.0 (1.2)%	45.8 (1.8)% **	37.5 (1.8)% ***	27.0 (0.7)%	20.3 (0.7)% ***	20.6 (1.7)% ***	17.1 (1.7)% ***
HDL-C meeting for Mets	30.3 (0.9)%	26.6 (1.0)% ***	24.7 (1.6)% *	21.0 (1.6)% ***	43.6 (0.7)%	36.8 (0.9)% ***	36.9 (2.0)% **	28.5 (2.0)% ***
Blood Pressure meeting for Mets	41.0 (1.0)%	39.2 (1.0)%	40.8 (1.1)%	40.1 (1.2)%	33.1 (0.7)%	29.3 (0.7)% ***	25.5 (0.8)% ***	23.4 (0.8)% ***
FPG meeting for Mets	48.2 (1.8)%	45.2 (1.9)% *	44.6 (1.5)%	44.4 (1.4)% *	35.1 (1.8)%	29.6 (2.0)% ***	29.2 (1.6)% **	23.6 (1.6)% ***
Metabolic Syndrome	35.8 (0.9)%	30.4 (1.0)% *	33.0 (1.7)% **	27.3 (1.7)% ***	29.2 (0.7)%	18.4 (0.8)% ***	21.5 (1.5)% ***	14.6 (1.5)% ***

Values are presented as weighted proportions (with standard errors). Abbreviations: NEG, none exercise group; APA, aerobic physical activity group; MSE, muscle strengthening exercise group; CEG, combined exercise group; Mets, metabolic syndrome; HDL-C, high-density lipoprotein cholesterol; FPG, fating plasma glucose. * *p*-value < 0.05, ** *p*-value < 0.01, *** *p*-value < 0.001.

### 3.3. ORs on the Prevalence of Metabolic Syndrome

The difference in the prevalence of metabolic syndrome according to meeting guidelines for APA and MSE. In both sexes, all other exercise groups had lower likelihood of diagnosis of metabolic syndrome compared to NEG in all models. Among men, ORs of APA, MSE, and CEG were 0.85 (95% CI, 0.74–0.98), 0.81 (95% CI, 0.67–0.99), and 0.65 (95% CI, 0.54–0.78) in model 3, respectively. Among women, ORs of APA, MSE, and CEG were 0.83 (95% CI, 0.73–0.93), 0.73 (95% CI, 0.58–0.91), and 0.74 (95% CI, 0.58–0.93) in model 3, respectively (Figure 1).

BMI, which was included as an adjustment variable in the main analysis, may have multicollinearity with waist circumference, a component of metabolic syndrome. However, excluding BMI from the model did not result in statistically significant differences in the main results.

### 3.4. Generalized Linear Model on Metabolic Components

Among men, CEG had statistically significantly lower levels of waist circumference and triglycerides and higher level of HDL-C compared to NEG after adjusting by age, lifestyle habits, BMI, and socioeconomic status. Also, APA had lower level of waist circumference and higher level of HDL-C and MSE had lower level of waist circumference.

Among women, CEG and APA had lower levels of waist circumference, triglycerides, and FPG and higher level of HDL-C than NEG. MSE also had lower levels of waist circumference, triglycerides, and SBP and higher level of HDL-C (Table 4).

## 4. Discussion

In this study, we found that meeting guidelines for APA and MSE was independently associated with a lower prevalence of metabolic syndrome and had favorable association with various metabolic syndrome components. Subjects who met both APA and MSE had the lowest prevalence of metabolic syndrome. APA and MSE also had likelihood of lower prevalence of metabolic syndrome compared to NEG. These findings were consistent with previous studies results that physical activity had beneficial effects on lowering risk of developing metabolic syndrome [23,24]. Among subjects with metabolic syndrome, APA, MSE and CEG improved health outcomes significantly [25]. Even though the relationship between MSE and metabolic syndrome was not consistent [26], we ascertained that MSE had an association with lower prevalence of metabolic syndrome.

In previous meta-analysis studies, APA was beneficial effects in reducing waist circumference, SBP and DBP and increasing HDL [27]. In subjects with metabolic syndrome, APA improved waist circumference, TG, DBP and FPG [13,26]. In obese subjects, APA was effective to manage metabolic syndrome, but not in MSE [16]. In this study, APA was associated to lower levels of waist circumference, triglycerides, FPG and higher level of HDL-C in women. In men, APA had relationship with lower level of waist circumference and higher level of HDL-C. However, significant relationship of APA with triglycerides and FPG was not found in men.

Unlike in the current study, MSE did not reduce body mass index and waist circumference in some studies [26,28], but other studies showed that MSE had beneficial effects on reducing waist circumference [14,29]. In impaired glucose tolerance and diabetic patients, MSE significantly reduced glycated hemoglobin [14,29]. For participants with Type 2DM or obesity, MSE significantly reduced SBP [14,30]. MSE had beneficial effects on managing blood pressure. However, in the general population, there was no beneficial effects on metabolic parameters [26]. In addition, metabolic effect of MSE was different according to intensity of exercise and age. In this study, MSE was associated to lower levels of waist circumference, SBP, triglycerides and higher level of HDL-C in women. In men, MSE had relationship with lower level of waist circumference.

In this study, CEG was associated with a reduced prevalence of metabolic syndrome, consistent with previous studies [13,26]. We found that CEG was associated with the lowest prevalence of metabolic syndrome. Among obese or overweight participants, CEG had no difference in reducing waist circumference compared with APA, but among normal participants, CEG had more beneficial effects [13,28]. As in the current study, APA and CEG were associated with decreased FPG in women, and there was no difference between APA and CEG [13,26]. In type 2 diabetic patients, APA and CEG were more effective in managing and preventing metabolic syndrome than MSE [15]. CEG had more beneficial effects in reducing glycated hemoglobin than other exercises groups [31]. APA and CEG were associated with increased HDL-C and decreased triglycerides [13,26,32]. Among healthy and young people, exercise did not affect cholesterol, but for old or obese people. APA and CEG were associated with reducing cholesterol [33].

Insulin resistance and chronic inflammation are essential mechanisms for the pathogenesis of metabolic syndrome [34]. APA is associated with higher energy expenditure than MSE during the same exercise period [29,35], reducing chronic inflammation [25], and increasing the expression of PGC-1a, which protects against mitochondrial disorders such as apoptosis or oxidative damage [36]. Increased skeletal muscle was associated with higher resting metabolic rate (RMR), as well as higher use of glycogen and fatty acids [29]. MSE is superior to aerobic exercise in increasing skeletal muscle mass. An increase of 1 kg muscle mass should result in an RMR increase of 21 kcal/kg of new muscle. Thus MSE elevates RMR [37]. In addition, APA and MSE are associated with increased glucose uptake and decreased visceral fat [25]. Different mechanisms mediate these metabolic changes. In CEG, we considered both mechanisms beneficial in managing metabolic syndrome.

Previous studies showed that aerobic exercise was more effective to decrease the prevalence of metabolic syndrome than MSE [15,16]. In previous studies, participants were overweight or obese [33] or had type 2 diabetes or metabolic syndrome [13,15]. In contrast, the participants of this study were the community dwelling population including normal weight subjects and subjects without type 2 diabetes or metabolic syndrome. In previous studies, aerobic exercise was defined as using the treadmill, jogging, or cycling [15,31]. We used the terms aerobic physical activity instead aerobic exercise to include a broader range of daily activities. We defined aerobic activity based on physical activity through work, recreation, or moving a location, including treadmill, jogging, or cycling. The metabolic effects of APA were consistent. Most of muscle-strengthening studies focused on machine-based weight training [32,37]; however, few studies include free weights or bodyweight exercises [15]. In this study, MSE included machine-based weight training, free weights, and body weight exercise.

Our study has several limitations. First, since this study was a cross-sectional study, the causality of physical activity and metabolic syndrome could not be drawn. Underlying diseases with metabolic syndrome could negatively affect physical activity even though subjects with cardiovascular diseases were excluded. Second, because this study was a secondary data analysis of national health surveys, there may be residual confounding. For example, diet quality and sleep duration could act as a confounder, but was not included in the analysis. Moreover, the subjects’ detailed history of drug intake was not investigated. Third, since self-reported questionnaires were used to assess physical activity, there may be reporting bias or misclassification. Therefore, we think that longitudinal or interventional studies are warranted to elaborate the association between meeting exercise guidelines and the prevalence of metabolic syndrome.

Exercise prescription is a cost-effective strategy to prevent and manage metabolic syndrome [38,39]. Furthermore, we found that all exercise groups have beneficial effect on managing metabolic syndrome and its components. In Korea, the National Health Promotion Plan was developed to improve aerobic physical activity; establishment of physical activity programs or creating a physical activity-friendly environment such as running road and bike path [17]. However, the proportion of Korean adults meeting MSE recommendation was still low and the proportion of those meeting APA has decreased [12,40]. Face-to-face interventions and counseling physical activity was effective to improve physical activity [12]. In Korean medical status, because of too many patients, there was not enough clinic hours to counsel physical activity. Despite insufficient clinic hours, clinicians counsel patients to meet exercise guidelines not only APA but also MSE and encourage combined exercise to improve metabolic health.

## 5. Conclusions

This study showed that meeting guidelines for APA and MSE was associated with lower prevalence of metabolic syndrome. Furthermore, CEG was associated with the lowest metabolic syndrome prevalence. We suggest that both APA and MSE should be educated and combined exercise should be encouraged to improve metabolic health.

## Figures and Tables

**Figure 1 nutrients-14-05391-f001:**
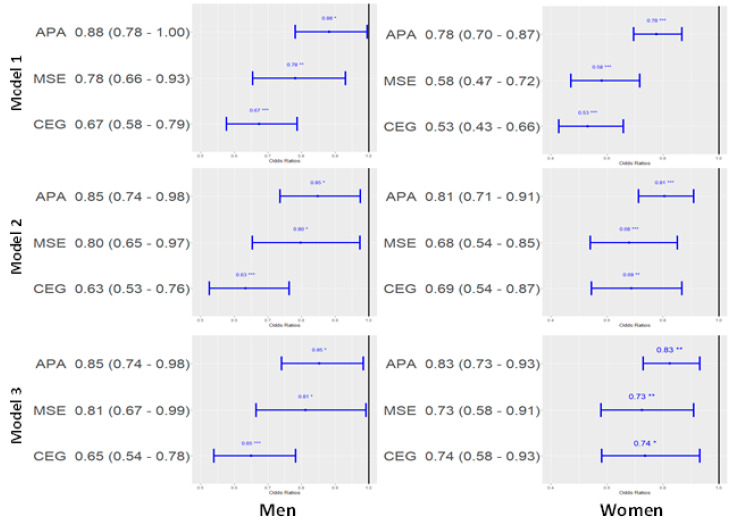
Odds ratio and 95% confidence interval of each exercise group on the prevalence of metabolic syndrome (reference: none exercise group). Model 1: adjusted by age (continuous), Model 2: adjusted by age (continued), smoking (present or not), alcohol (<1/year, <1/month, ≥2/month), BMI (continuous), Model 3: adjusted by age (continuous), smoking (present or not), alcohol (<1/year, <1/month, ≥2/month), BMI (continuous), locality of dwelling (urban or rural), education level (under elementary school, elementary school graduate or middle school less, middle school graduate or high school less, university undergraduate or graduate) and household income (lower, middle, middle upper, and upper). Abbreviations: APA, aerobic physical activity group; MSE, muscle strengthening exercise group; CEG, combined exercise group.* *p*-value < 0.05, ** *p*-value < 0.01, *** *p*-value < 0.001.

**Table 1 nutrients-14-05391-t001:** General characteristics of study participants.

	Men (Unweighted *n* = 9670)	Women (Unweighted *n* = 12,797)	*p*-Value
Age	55.3 (10.5)	56.6 (11.2)	<0.001
Body mass index	24.5 (3.1)	23.8 (3.4)	<0.001
Waist Circumference	86.9 (8.4)	80.4 (9.3)	<0.001
Total Cholesterol	195.0 (37.0)	198.5 (37.1)	<0.001
Triglyceride	173.5 (146.6)	121.5 (82.4)	<0.001
HDL-C	47.1 (11.3)	54.0 (12.7)	<0.001
LDL-C	116.9 (33.9)	119.9 (33.4)	0.002
FPG	106.4 (26.3)	100.3 (21.9)	<0.001
Hemoglobin A1c	5.9 (0.9)	5.8 (0.8)	<0.001
Hypertension	38.9 (0.6)%	32.9 (0.6)%	<0.001
Diabetes Mellitus	16.5 (0.4)%	12.0 (0.4)%	<0.001
Hypercholesterolemia	21.6 (0.5)%	28.3 (0.5)%	<0.001
Metabolic syndrome	33.0 (0.6)%	24.8 (0.5)%	<0.001
Exercise groups			<0.001
NEG	42.4 (0.6)%	52.6 (0.6)%	
APA	29.7 (0.6)%	33.5 (0.5)%	
MSE	10.6 (0.4)%	5.7 (0.3)%	
CEG	17.3 (0.5)%	8.2 (0.3)%	

Values are presented as weighted means (with standard errors) for continuous variables or weighted proportions (with standard errors) for categorical variables, respectively. Abbreviations: HDL-C, high-density lipoprotein cholesterol; LDL-C, low-density lipoprotein cholesterol; FPG, fasting plasma glucose; NEG, none exercise group; APA, aerobic physical activity; MSE, muscle strengthening exercise; CEG, combined exercise group.

**Table 2 nutrients-14-05391-t002:** Metabolic profiles of study participants according to exercise groups.

	Men				Women			
	NEG (Unweighted *n* = 3885, Ref)	APA (Unweighted *n* = 2518)	MSE (Unweighted *n* = 971)	CEG (Unweighted *n* = 1486)	NEG (Unweighted *n* = 6624, Ref)	APA (Unweighted *n* = 3879)	MSE (Unweighted *n* = 694)	CEG (Unweighted *n* = 943)
Age (year)	55.7 (10.9)	57.6 (10.4) ***	54.1 (10.3) **	55.0 (9.9) ***	58.1 (11.8)	56.6 (10.3) ***	54.9 (10.4) **	53.1 (9.2) ***
BMI (kg/m^2^)	24.4 (3.2)	24.3 (2.7)	24.6 (3.1)	24.5 (2.7)	24.0 (3.5)	23.3 (3.0) **	23.7 (3.3) ***	22.9 (3.0) ***
WC (cm)	87.2 (9.0)	86.7 (7.7)	86.9 (8.3)	86.0 (7.7) ***	81.4 (9.6)	79.0 (8.4) ***	80.0 (9.0) ***	77.3 (8.0) ***
Triglyceride (mg/dL)	179.7 (154.6)	165.8 (134.7)	178.1 (148.6) *	155.0 (127.3) ***	128.3 (88.0)	113.4 (73.7) ***	115.5 (73.4) ***	108.1 (81.4) ***
HDL-C (mg/dL)	46.2 (11.3)	47.0 (11.1) ***	47.6 (11.2)	48.6 (11.1) ***	52.8 (12.4)	55.4 (13.9) ***	54.6 (12.5) ***	57.9 (13.3) ***
FPG (mmol/L)	107.3 (26.8)	105.1 (24.3)	106.0 (26.1)	105.7 (26.4)	101.9 (23.7)	98.3 (20.8) ***	99.2 (20.0) ***	96.5 (16.6) ***
SBP (mmHg)	122.1 (15.2)	122.6 (14.9) *	121.1 (14.9)	122.1 (15.4)	120.7 (17.9)	117.2 (16.8) ***	118.6 (17.3) ***	115.5 (16.5) ***
DBP (mmHg)	79.2 (10.2)	78.5 (10.2) *	79.4 (10.2)	79.3 (9.8)	74.8 (9.9)	74.0 (9.0) *	75.3 (9.4)	74.5 (9.3)
Hypertension (%)	40.20 (0.9)%	43.50 (1.1)% *	36.90 (1.9)%	36.30 (1.9)% *	37.80 (0.7)%	26.00 (0.9)% ***	28.90 (1.9)% ***	22.00 (1.9)% ***
Hypercholesterolemia (%)	22.00 (0.8)%	22.00 (1.0)%	24.00 (1.6)%	18.20 (1.1)% **	29.50 (0.7)%	26.60 (0.8)% **	29.20 (2.0)%	25.90 (1.76)% *
Diabetes Mellitus (%)	18.30 (0.7)%	15.60 (0.8)% **	15.10 (1.3)%	15.30 (1.3)% *	14.20 (0.5)%	8.60 (0.6)% ***	10.30 (1.5)% ***	6.80 (1.5)% ***

* *p*-value < 0.05, ** *p*-value < 0.01, *** *p*-value < 0.001. Values are presented as weighted means (with standard errors) for continuous variables or weighted proportions (with standard errors) for categorical variables, respectively. Abbreviations: NEG, none exercise group; APA, aerobic physical activity group; MSE, muscle strengthening exercise group; CEG, combined exercise group; BMI, body mass index; WC, waist circumference; HDL-C, high-density lipoprotein cholesterol; FPG, fating plasma glucose; SBP, systolic blood pressure; DBP, diastolic blood pressure.

**Table 4 nutrients-14-05391-t004:** Generalized linear model of each exercise group on metabolic components (reference: NEG).

Model 1						
	WC	SBP	DBP	TG	HDL-C	FPG
Men						
APA	−0.29 [−0.78, 0.20]	−0.41 [−1.27, 0.46]	−0.27 [−0.84, 0.29]	−4.75 [−14.83, 5.32]	1.46 *** [0.82, 2.09]	−1.06 [−2.50, 0.39]
MSE	−0.55 [−1.15, 0.06]	−0.09 [−1.25, 1.07]	−0.1 [−0.88, 0.67]	−10.19 [−22.57, 2.20]	0.87 [−0.02, 1.76]	−2.52 * [−4.77, −0.27]
CEG	−1.24 *** [−1.77, −0.71]	0.26 [−0.84, 1.35]	−0.06 [−0.74, 0.62]	−26.15 *** [−36.57, −5.73]	2.38 *** [1.61, 3.15]	−1.45 [−3.33, 0.43]
Women						
APA	−0.67 ** [−1.10, −0.24]	0.01 [−0.70, 0.71]	0.33 [−0.11, 0.76]	−10.42 *** [−14.01, −6.83]	1.21 *** [0.65, 1.76]	−1.62 *** [−2.57, −0.67]
MSE	−2.05 *** [−2.78, −1.33]	−2.48 *** [−3.83, −1.13]	−0.88 * [−1.68, −0.07]	−13.81 *** [−20.32, −7.30]	2.29 *** [1.06, 3.52]	−3.10 *** [−4.94, −1.26]
CEG	−2.93 *** [−3.54, −2.32]	−1.76 ** [−2.99, −0.53]	−0.52 [−1.30, 0.27]	−16.40 *** [−22.91, −9.89]	4.11 *** [3.09, 5.12]	−3.68 *** [−5.05, −2.31]
Model 2						
	WC	SBP	DBP	TG	HDL-C	FPG
Men						
APA	−0.54 *** [−0.79, −0.28]	−0.55 [−1.40, 0.31]	−0.5 [−1.06, 0.06]	−3.85 [−14.02, 6.32]	1.33 *** [0.71, 1.95]	−0.97 [−2.42, 0.49]
MSE	−0.52 ** [−0.85, −0.20]	−0.17 [−1.33, 1.00]	−0.31 [−1.10, 0.47]	−7.13 [−19.63, 5.36]	0.52 [−0.36, 1.41]	−2.02 [−4.32, 0.27]
CEG	−1.47 *** [−1.76, −1.18]	0.01 [−1.10, 1.12]	−0.44 [−1.12, 0.25]	−23.61 *** [−34.43, −2.79]	1.87 *** [1.14, 2.60]	−1.24 [−3.15, 0.67]
Women						
APA	−0.31 ** [−0.53, −0.09]	0.14 [−0.56, 0.84]	0.42 [−0.02, 0.85]	−8.82 *** [−12.32, −5.32]	0.96 *** [0.42, 1.50]	−1.38 ** [−2.32, −0.43]
MSE	−0.47 * [−0.84, −0.11]	−1.85 ** [−3.17, −0.54]	−0.5 [−1.28, 0.28]	−9.66 ** [−16.06, −3.27]	1.64 ** [0.44, 2.85]	−2.10 * [−3.90, −0.30]
CEG	−0.82 *** [−1.13, −0.51]	−1.06 [−2.27, 0.15]	−0.06 [−0.83, 0.71]	−10.74 *** [−17.04, −4.44]	3.11 *** [2.13, 4.09]	−2.65 *** [−3.84, −1.46]
Model 3						
	WC	SBP	DBP	TG	HDL-C	FPG
Men						
APA	−0.60 *** [−0.85, −0.34]	−0.42 [−1.28, 0.44]	−0.55 [−1.10, 0.01]	−3.24 [−13.61, 7.13]	1.38 *** [0.76, 2.00]	−0.74 [−2.20, 0.73]
MSE	−0.58 *** [−0.91, −0.25]	0.07 [−1.11, 1.24]	−0.39 [−1.18, 0.40]	−5.23 [−17.65, 7.19]	0.46 [−0.43, 1.34]	−1.61 [−3.93, 0.71]
CEG	−1.61 *** [−1.90, −1.32]	0.44 [−0.68, 1.56]	−0.57 [−1.26, 0.12]	−20.55 *** [−31.24, −9.86]	1.90 *** [1.16, 2.65]	−0.49 [−2.45, 1.47]
Women						
APA	−0.29 ** [−0.50, −0.07]	0.2 [−0.50, 0.90]	0.37 [−0.06, 0.80]	−8.32 *** [−11.80, −4.84]	0.86 ** [0.32, 1.40]	−1.29 ** [−2.23, −0.34]
MSE	−0.44 * [−0.80, −0.07]	−1.59 * [−2.90, −0.28]	−0.48 [−1.26, 0.30]	−8.78 ** [−15.23, −2.33]	1.48 * [0.27, 2.69]	−1.82 [−3.67, 0.03]
CEG	−0.73 *** [−1.04, −0.42]	−0.82 [−2.03, 0.40]	−0.09 [−0.86, 0.69]	−9.87 ** [−16.28, −3.46]	2.84 *** [1.85, 3.83]	−2.26 *** [−3.45, −1.07]

* *p*-value < 0.05, ** *p*-value < 0.01, *** *p*-value < 0.001, Values are presented as odds ratios (95% confidence intervals). Model 1 adjusted for age (continuous), Model 2 adjusted for age (continued), smoking (present or not), alcohol (<1/year, <1/month, ≥2/month), BMI (continuous). Model 3 adjusted for age (continuous), smoking (present or not), alcohol (<1/year, <1/month, ≥2/month), BMI (continuous), locality of dwelling (urban or rural), education level (under elementary school, elementary school graduate or middle school less, middle school graduate or high school less, university undergraduate or graduate) and household income (lower, middle, middle upper, and upper). Abbreviations: WC, waist circumference; SBP, systolic blood pressure; DBP, diastolic blood pressure; TG, triglycerides; HDL-C, high-density lipoprotein cholesterol; FPG, fating plasma glucose; NEG, non-exercise group; APA, aerobic physical activity group; MSE, muscle strengthening exercise group; CEG, combined exercise group.

## Data Availability

The datasets used and/or analyzed during the current study are available from the Korea National Health and Nutrition Examination Survey (KNHANES) official website.at https://knhanes.kdca.go.kr/knhanes/eng/index.do (accessed on 1 April 2021).

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
