# Peer review of "The Difference in the Prevalence of Metabolic Syndrome According to Meeting Guidelines for Aerobic Physical Activity and Muscle-Strengthening Exercise: A Cross-Sectional Study Performed Using the Korea National Health and Nutrition Examination Survey, 2014–2019"

_nutrients, 2022, doi:10.3390/nu14245391_

Round 1
Reviewer 1 Report
Dear Authors,
However the design of the study and statistical model are correct, the manuscript should be thorougly rewritten. Please follow the instruction for the authors for the references.
Best regards
Author Response
I am grateful for reviewing our study. Also, I am thankful for your kind comments and helping to make better study. I edited based on reviewer’s opinions.
We review this study entirely and revised according to author’s guidelines.
We uploaded revised file.

Reviewer 2 Report
The authors used a cross-sectional study from Korea to examine the association between physical exercise and metabolic syndrome. The findings were relevant to understanding the health effect of exercise. I hope the comments below could help strengthen the paper.
The cross-sectional design of the study precludes causal inference. The authors should acknowledge this explicitly in the Limitations section and refrain from use of causal languages throughout the manuscript such as “reduce”, “affect”, and “effect”.
The Discussion section needs to discuss the potential causes of disparities of findings from previous studies. Was it due to different design, sample, measurement, or some latent divergences?
The Discussion section should also include policy implications of the findings.
Was there a Wald test to confirm that the odds ratio of CEG was greater in magnitude than that of MSE and APA (line 188)? If not, then the claim about CEG being “most” or “more” effective was unfounded.
The “dose-response” association was not a limitation (line 275).
The paper needs serious copy-editing and proofreading by professionals. There are too many grammar and syntax errors to allow proper interpretation of the text.
Author Response
I am grateful for reviewing our study. Also, I am thankful for your kind comments and helping to make better study. I edited based on reviewer’s opinions.
1)The authors used a cross-sectional study from Korea to examine the association between physical exercise and metabolic syndrome. The findings were relevant to understanding the health effect of exercise. I hope the comments below could help strengthen the paper.
: We added the findings and comments to strengthen the paper.
2)The cross-sectional design of the study precludes causal inference. The authors should acknowledge this explicitly in the Limitations section and refrain from use of causal languages throughout the manuscript such as “reduce”, “affect”, and “effect”.
: We acknowledge the limitation of cross-sectional design of this study and refrain using casual words such as “reduce”, “affect” and “effect”, instead using "association", "higher" and "lower".
3)The Discussion section needs to discuss the potential causes of disparities of findings from previous studies. Was it due to different design, sample, measurement, or some latent divergences?
: These results could be explained in that previous studies participants were overweight or obesity or with type 2 diabetes or metabolic syndrome. But, in this study, we included participants with normal weight or without type 2 diabetes or metabolic syndrome. In previous study, aerobic exercise was defined performing treadmill, jogging, or cycling. We defined aerobic activity was based on physical activity through work, recreation or moving a location including treadmill, jogging, or cycling. Effects of APA was consistent. Most of muscle strengthening exercise was defined machine-based weight training, few studies include free weights or bodyweight exercises. In this study, we defined MSE including machine-based weight training, free weights, and body weight exercise. Also, most previous studies have been conducted in Europe or America.
4) The Discussion section should also include policy implications of the findings.
: We added health policy to decrease the prevalence of metabolic syndrome.
But the proportion of meeting MSE was low, and the proportion of meeting APA has decreased. We need health policies to encourage following exercise guidelines such as APA or MSE.
5) Was there a Wald test to confirm that the odds ratio of CEG was greater in magnitude than that of MSE and APA (line 188)? If not, then the claim about CEG being “most” or “more” effective was unfounded.
: We do not analyze directly between the prevalence of each exercise guidelines. So, we delete above sentence.
6)The “dose-response” association was not a limitation (line 275).
: We accept reviewer's opinion, so we delete above sentence.
7) The paper needs serious copy-editing and proofreading by professionals. There are too many grammar and syntax errors to allow proper interpretation of the text.
: We got copy-editing and proofreading to correct grammar and errors by professional editor. We uploaded the certificate.

Reviewer 3 Report
General comment:
The authors have used the Korea National Health and Nutrition Examination Survey (KNHANES) 2014-2019 to investigate correlations between the prevalence of metabolic syndrome and meeting the aerobic physical activity, muscle strengthening exercise, and combined exercise group guidelines compared with a non-exercise group. They conclude that aerobic physical activity and muscle strengthening are associated to a reduce the prevalence of metabolic syndrome independently. Combined exercise was the most effective strategy. “Exercise was more beneficial in females in managing metabolic syndrome components”.
Detailed comments
1. ABS, lines 13-14: The sentence “However, the evidence on the benefits of physical activity in preventing metabolic syndrome is limited to apply in Korean population” is unclear. What do the authors mean by that?
2. Please include in the ABS the population sample characteristics.
3. Please define clearly what do you mean by NEG: Subjects not exercising all the way, or not meeting the recommendations…
4. Key abbreviations such as APA, MSE, CEG and NEG are only defined in the Abs but nowhere in the text. They should be clearly defined on top of the MS as well as recalled somewhere also in the text.
5. Ln 71: “There is an increasing the prevalence of,,,”. Correct sentence please.
6. Lns 73-74: Were the subjects excluded for the missing data randomly distributed? I.e. was there any excess prevalence in other key variables in the missing data population?
7. Ln 87: the sentences should be: “…high blood pressure of >130/85 for SBP or DBP…”
8. Make lns 87-88 and 90-91 consistent.
9. Results: data presentation from Ln 150 is confusing, no clearly-presented.
10. Table 1: what do the p values indicate? i.e. differences between sexes or else?
11. Table 1: the p value for the prevalence of diabetes between sexes is misplaced, that of the MS too..
12. Model 2 contains one variable (BMI), that may closely related to waist circumference. How could such a parameter proximity affect the results?
13. Ref. 19 reports another study in Korea population going into the same direction of the present one. This detracts on the originality of the present one.
14. Lns 221-226: could the positive results observed in female vs male subjects be associated to the hormonal status?
15. Ln 262: “A previous study showed that aerobic exercise was more effective than MSE”. In what?
16. Contrary to what stated in lines 278-279, there are other published studies on this issue (see: PMID: 29356607; PMID: 30755271; PMID: 24389523; PMID: 32342447 (Review); PMID: 27773709 (Review); PMID: 28622914; etc).
The last sentence in the conclusion (lns 290-291) seems to go beyond the limitation of this study (a cross-sectional-observational, not an interventional one).
Author Response
1) ABS, lines 13-14: The sentence “However, the evidence on the benefits of physical activity in preventing metabolic syndrome is limited to apply in Korean population” is unclear. What do the authors mean by that?
: In ABS, we deleted this sentence, instead these sentences appeared at introduction. We mean that there has been insufficient evidence on the benefits of physical activity in preventing metabolic syndrome in Korean, because previous studies participants were European and American.
2) Please include in the ABS the population sample characteristics.
: We add "This was a cross-sectional study on 22,467 Korean adults aged 40 years or older using data from the Korea National Health and Nutrition Examination Survey, 2014-2019."
3) Please define clearly what do you mean by NEG: Subjects not exercising all the way, or not meeting the recommendations.
: It is necessary to display 4 groups clearly, so we add the below sentences.
The study participants were categorized into four groups; 1) none exercise group (NEG), 2) meeting only APA, 3) meeting only MSE, and 4) combined exercise group (CEG).
4) Key abbreviations such as APA, MSE, CEG and NEG are only defined in the Abs but nowhere in the text. They should be clearly defined on top of the MS as well as recalled somewhere also in the text.
: We revised it.
5) Ln 71: “There is an increasing the prevalence of,,,”. Correct sentence please.
: We added sentences to mean clearly,
"we enrolled study subjects over 40 years old as the prevalence of metabolic syndrome increased in subjects over 40 years"
6)Lns 73-74: Were the subjects excluded for the missing data randomly distributed? I.e. was there any excess prevalence in other key variables in the missing data population?
: Compared with subjects regardless of whether participating or not in physical activity survey, there was no statistical difference in the prevalence of metabolic syndrome and the components of metabolic syndrome in subjects participating physical activity survey group.
7) Ln 87: the sentences should be: “…high blood pressure of >130/85 for SBP or DBP…”
: We change the above sentence
Systolic blood pressure (SBP)≥130 mmHg or diastolic blood pressure (DBP)≥85 mmHg
8) Make lns 87-88 and 90-91 consistent.
We also change it. You can see it at method section in uploaded file.
9) Results: data presentation from Ln 150 is confusing, no clearly-presented.
: We change sentence to mean clearly. You can also see it at results section in uploaded file.
A total of 22,467 subjects, male subjects were 43.0%.The mean age of males and females was 55.3±10.5 years and 56.6±11.2 years, respectively. Males had a higher prevalence of metabolic syndrome than females (33.0±0.6% VS. 24.8±0.5%, p<0.001). The proportions of each exercise group were 42.4±0.6% for NEG, 10.6±0.4% for MSE, 29.7±0.6% for APA and 17.3±0.5% for CEG in males and 52.6±0.6% for NEG, 5.7±0.3% for MSE, 33.5±0.5% for APA and 8.2±0.3% for CEG in females.
10)Table 1: what do the p values indicate? i.e. differences between sexes or else?
We were intended to show the difference between sexes. But, table 1 was about general characteristics of subjects, so we deleted p-value.
11) Table 1: the p value for the prevalence of diabetes between sexes is misplaced, that of the MS too..
: We deleted it.
12) Model 2 contains one variable (BMI), that may closely related to waist circumference. How could such a parameter proximity affect the results?
: We presented p-value of general logistic regression in model 2 and in model 3 with or without BMI. There was difference including BMI or not, but there was no statistical difference in p-value. Among males, meeting guideline for APA there was changes in the p-value.
|
Man |
Model 2 |
p-value |
Model 2 w/o BMI |
p-value |
|
|
APA |
0.0211 * |
APA |
0.06355 |
|
|
MSE |
0.0264* |
MSE |
0.01364 * |
|
|
CEG |
1.72e-06 *** |
CEG |
1.94e-06 *** |
|
|
Model 3 |
p-value |
Model 3 w/o BMI |
p-value |
|
|
APA |
0.02883 * |
APA |
0.08471 |
|
|
MSE |
0.04152 * |
MSE |
0.02866 * |
|
|
CEG |
5.82e-06 *** |
CEG |
6.69e-06 *** |
|
|
|
|
|
|
|
Woman |
Model 2 |
p-value |
Model 2 w/o BMI |
p-value |
|
|
APA |
0.000502 *** |
APA |
3.02e-05 *** |
|
|
MSE |
0.000864 *** |
MSE |
7.47e-07 *** |
|
|
CEG |
0.001677 ** |
CEG |
1.20e-08 *** |
|
|
Model 3 |
p-value |
Model 3 w/o BMI |
p-value |
|
|
APA |
0.00212 ** |
APA |
0.000400 *** |
|
|
MSE |
0.00559 ** |
MSE |
4.34e-05 *** |
|
|
CEG |
0.01119 * |
CEG |
7.91e-06 *** |
13) Ref. 19 reports another study in Korea population going into the same direction of the present one. This detracts on the originality of the present one.
: Previous study investigated the association between intensity of aerobic exercise, strength exercise and flexibility exercise and the prevalence of metabolic syndrome. We exam the prevalence of metabolic syndrome according to meeting exercise guidelines. In addition, this study analyzed the combined effect of aerobic exercise and strength exercise together.
14) Lns 221-226: could the positive results observed in female vs male subjects be associated to the hormonal status?
: According to table 2, among females there was an age difference statistically each exercise guideline groups, but not in males. Also, among females, the proportion of meeting MSE and CEG was lower than other groups. We think different age was the factor to explain positive results in female.
15) Ln 262: “A previous study showed that aerobic exercise was more effective than MSE”. In what?
: We added sentences to mean clearly.
"A previous study showed that aerobic exercise was more effective to decrease the prevalence of metabolic syndrome than MSE"
16)Contrary to what stated in lines 278-279, there are other published studies on this issue (see: PMID: 29356607; PMID: 30755271; PMID: 24389523; PMID: 32342447 (Review); PMID: 27773709 (Review); PMID: 28622914; etc).
: Previous studies investigated the effects of aerobic or strength training in overweight, obese, diabetic or metabolic syndrome patients. This study aimed to investigate association between the meeting exercise guidelines and the prevalence of metabolic syndrome including general population with or without obese, overweight or diabetic.
17)The last sentence in the conclusion (lns 290-291) seems to go beyond the limitation of this study (a cross-sectional-observational, not an interventional one).
: According to reviewer's opinion, we delete this sentence.

Round 2
Reviewer 2 Report
The policy discussion in the end needs to be fleshed out. What policies have been in place? What policies can be improved?
There are still plenty of grammar and syntax errors. For instance, in the abstract, it should be "participated in" instead of "conducted in".
Author Response
I am thankful that your review is very helpful and make the article better.
- The policy discussion in the end needs to be fleshed out. What policies have been in place? What policies can be improved?
: We added the below sentence at Ln306
"In Korea, the Ministry of Health and Welfare and the Institute for Health Promotion developed the National Health Promotion Plan to improve physical activity; establishment of physical activity programs, development and provision of individualized services to encourage physical activity, and creating a physical activity-friendly environment. However, the proportion of meeting MSE was still low and the proportion of meeting APA has decreased. Patient should be educated appropriate exercise protocol and encouraged following exercise guidelines. It is necessary to establish institutional systems to counsel exercise appropriately in medical environment. "
There are several strategies to promote physical activity: 1) establishment of physical activity program 2) establishment of infrastructures on community basis 3) presenting evidence about benefits of physical activity and harm of sedentary lifestyle 4) development individualized services 5) building a physical-friendly environment 6) campaign to encourage exercise
ref) Ministry of Health and Welfare. Integrated Health Promotion Program [Internet]. Seoul: Korea Health Promotion Institute; 2021 [cited 2022 Mar 27]. Available from: https://www.khealth.or.kr/board?menuId=M ENU00829&siteId=null
2. There are still plenty of grammar and syntax errors. For instance, in the abstract, it should be "participated in" instead of "conducted in".
: We request professional editor at Editage to ensure language and grammar accuracy. We upload certification of Editage.

Reviewer 3 Report
Comments to authors' reply:
It doesn't seem that the requested, following sentence has been added to the MS: "Compared with subjects regardless of whether participating or not in physical activity survey, there was no statistical difference in the prevalence of metabolic syndrome and the components of metabolic syndrome in subjects participating physical activity survey group."
Line 81: ...please correct: Inferioir
Lines 82-83: The authors did not reply properly to my question, I meant to comment about the similarity in the MS criteria between AHA ot of NHLBI. Furthermore, (current) Ref 16 does not report the criteria neither of AHA nor of NHLBI, rather those of WHO 99, NCEP and IDF.
LIne 85-87: ..."We defined diabetes as (FBS)≥126mg/dL, a glycated hemoglobin level ≥6.5%, taking oral diabetes medications or using insulin therapy" Do you mean, using either one or more than one of these criteria together?.
#9 reply to my comment: I can't access the "uploaded file". So, were the males and females groups statistically different?
#12 reply to my comment: The reported sentence:... "We presented p-value of general logistic regression in model 2 and in model 3 with or without BMI. There was difference including BMI or not, but there was no statistical difference in p-value. Among males, meeting guideline for APA there was changes in the p-value"... should be placed in the text too.
#13 reply to my comment: The results of the previous and of the current studies seem to be the same, although achieved from opposite starting points (apart from the here-analyzed combined effect of aerobic exercise and strength exercise together). This limitatiion markedly detracts from the originality of the present MS.
#14 reply to my comment: The authors here state that: "According to table 2, among females there was an age difference statistically each exercise guideline groups, but not in males. Also, among females, the proportion of meeting MSE and CEG was lower than other groups. We think different age was the factor to explain positive results in female." While this sentence is confusing and poorly written, it does't answer to my question.
Author Response
We are appreciated your review. It makes the article better.
We uploaded word file for answering your review.
Thank you.
